# QuickVideo: Real-Time Long Video Understanding with System Algorithm Co-Design

**Benjamin Schneider\***
*University of Waterloo*

*Benjamin.Schneider@uwaterloo.ca*

**Dongfu Jiang\***
*University of Waterloo*

*Dongfu.Jiang@uwaterloo.ca*

**Chao Du**
*SeaAI Lab*

*duchao@sea.com*

**Tianyu Pang**
*SeaAI Lab*

*tianyupang@sea.com*

**Wenhu Chen**
*University of Waterloo*

*Wenhu.Chen@uwaterloo.ca*

**Reviewed on OpenReview:** *https://openreview.net/forum?id=Rpcxgzcsuc*

## Abstract

Long video understanding has emerged as a crucial capability in real-world applications such as meeting summarization, video surveillance, educational lecture analysis, and content moderation. However, it remains computationally prohibitive for VideoLLMs, primarily due to two bottlenecks: 1) *sequential video decoding*, the process of converting the raw bit stream to RGB frames can take up to a minute for hour-long video inputs, and 2) *costly prefilling of up to several million tokens* for LLM inference, resulting in high latency and memory use. To address these challenges, we propose **QuickVideo**, a *system-algorithm co-design* that substantially accelerates long video understanding to support real-time downstream applications. It comprises three key innovations: **QuickCodec**, a parallelized CPU-based video decoder that achieves 2–3× speedup by splitting videos into keyframe-aligned intervals processed concurrently. **QuickPrefill**, a memory-efficient prefilling method using KV-cache pruning to support more frames with less GPU memory; and **an overlapping scheme** that overlaps CPU video decoding with GPU inference. Together, these components reduce the time required to process a long video input by a minute, enabling fast, efficient video understanding even on limited hardware. Experiments show that QuickVideo generalizes across durations and sampling rates, making long video processing feasible in practice.

## 1 Introduction

Video data has become the dominant modality for conveying information online.

As of 2023, video data accounts for two thirds of all data transmitted over the Internet (Su et al., 2024). Much of this data is "long video" ranging from minutes to hours in duration, from online conferencing, gaming, social networking, and movie streaming. This torrent of online video data demands efficient and automated understanding for problems such as content moderation (Akyon & Temizel, 2022), real-time surveillance (Yuan et al., 2023), and accessibility (Liu et al., 2021). Video Large Language Models (VideoLLMs) (Bai et al., 2025; Zhu et al., 2025a; Chen et al., 2025a) have emerged as powerful tools to support these downstream

---

* The first two authors have equal contribution

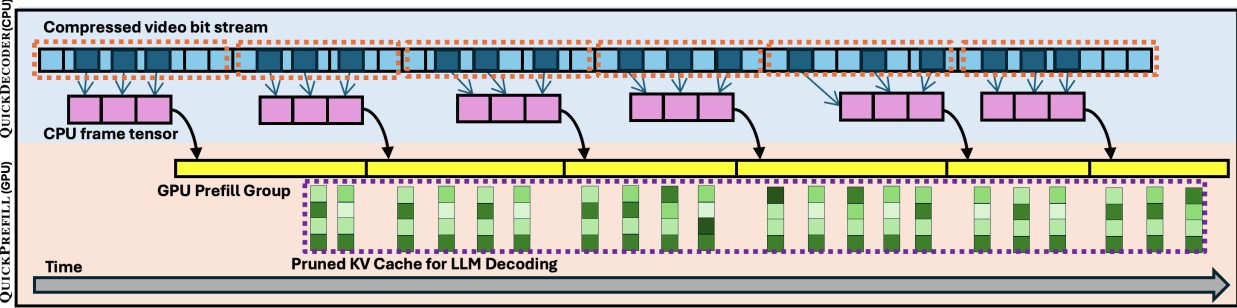

Figure 1: An overview of how QUICKVIDEO overlaps video decoding on CPU (QUICKCODEC) and prefill on GPU (QUICKPREFILL). QUICKCODEC concurrently processes intervals of the compressed video bit stream. QUICKPREFILL uses independent groups of frames, therefore it can begin prefill once the first frames are decoded, outputting carefully selected KV vectors. As QUICKCODEC loads frames synchronously, QUICKPREFILL can process the next prefill group immediately. This results in video decoding and prefill being almost enirely overlapped.

tasks. By natively processing entire video inputs, VideoLLMs exhibit phenomenal potential to understand and reason about video content, offering a practical solution for managing and extracting information from the exponentially growing flood of video data across the Internet (Zhou et al., 2024b).

However, using VideoLLMs for long video understanding suffers from several efficiency challenges. First, the entire video must be decoded from raw bitstreams into RGB frames before the model can begin processing. Current frameworks require up to a minute to decode the frames from an hour-long video input, introducing high latency before any context prefill can start. Second, the prefilling step itself is both computationally and memory intensive (Weng et al., 2024). Each frame—representing an instantaneous moment—can consume hundreds of tokens in the model context (Zhu et al., 2025b; Chen et al., 2025b). As a result, even a modest frame rate (e.g., 2 FPS) for an hour-long video can lead to millions of tokens, far exceeding the memory budget of standard GPUs. Qwen2.5-VL (Bai et al., 2025) introduced several architecture modification to accelerate video processing. However, using Qwen2.5-VL-7B, prefilling an hour-long HD video sampled at its native 2 fps still requires more than the 80 GB of memory offered by an A100/H100 SXM4 GPU. Even after reducing the frame sampling rate by 4×, prefilling still takes over 25 seconds on datacenter-grade hardware. These inefficiencies result in a frustrating user experience, characterized by long delays and prohibitively high hardware requirements. Users with limited computational resources are effectively excluded from accessing the long video understanding capabilities of VideoLLMs.

To mitigate the computational overhead of long video VideoLLMs use *extremely low frame sampling rates* when processing long video inputs, instead of their native 1-2 FPS Bai et al. (2025); Chen et al. (2025a). Frames are sampled as much as a minute apart during hour-long video understanding (Zhu et al., 2025b; Chen et al., 2025b). A minute gap between sampled frames can result in missing crucial video segments required for an understanding task. Low frame sampling rates also make fine-grained temporal and motion understanding impossible, as intervening frames are mostly removed (Nie et al., 2024). Effective long video understanding thus requires loading and prefilling thousands of frames while preserving temporal continuity. Developing faster, more efficient VideoLLMs is critical for enabling comprehension of videos that span hours.

Currently, video decoding and context prefilling are treated as disjoint and sequential stages in the VideoLLM pipeline. Moreover, video decoding is largely overlooked, despite contributing substantially to end-to-end latency. To remedy this, we introduce QUICKVIDEO, a framework for faster, memory-efficient long video understanding. QUICKVIDEO reduces the latency and resource requirements of these key bottlenecks in long video understanding. Our framework empowers fast video understanding on video inputs consisting of hundreds of thousands of frames, while maintaining the sampling rates required for fine-grained understanding. QUICKVIDEO introduces three core contributions for accelerating long video understanding in VideoLLMs:

**(1) System-Level → QUICKCODEC**: a drop-in replacement video decoder designed for VideoLLMs. By redesigning video decoding for VideoLLM frame sampling, we achieve a 2-3x speedup compared to existing

libraries when loading hour-long video inputs.

**(2) Algorithm-Level → QUICKPREFILL**: a group-based prefilling strategy combined with key-value (KV) cache pruning, which significantly reduces both computation time and memory usage during the prefilling stage, while incurring less than 3% accuracy degradation in most benchmarks.

**(3) Co-Design → Overlapped Execution Scheme**: the strategy tightly couples CPU-based QUICKCODEC and GPU-based QUICKPREFILL, enabling near-complete overlap to maximize efficiency. QUICKVIDEO reduces the time to infer a 30 minute video input by more than 3x, from 69.7 seconds to only 20.0 seconds. The results demonstrate the effectiveness of our system-algorithm co-design.

## 2 Background

We provide an overview of VideoLLM inference and key concepts in video processing. Although details vary, this background is broadly applicable to standard VideoLLM architectures and video standards. For clarity, we use "video decoding" to describe the process of decoding the compressed video into a tensor of video frames, and use "LLM decoding" to denote the process of auto-regressive decoding of a large language model.

### 2.1 VideoLLM Inference

VideoLLMs must first decode a compressed video into a packed frame tensor before tokenization. The resulting raw frames are then passed through a visual encoder, which converts them into video tokens suitable for input to the LLM. Unlike text preprocessing, which relies on lightweight tokenizers, video decoding is inherently slow on both CPU and GPU due to its sequential nature (Wiegand et al., 2003; Sullivan et al., 2012). Despite this, prior work in LLM video understanding has largely overlooked the latency incurred by this stage. Following preprocessing, the generation process of a VideoLLM consists of two stages: **(1) Prefill**, where both video and text tokens are processed to compute key-value (KV) caches for each transformer layer; and **(2) LLM decoding**, where tokens are generated autoregressively using the stored KV representations. The prefill stage is computationally expensive due to the quadratic complexity $\mathcal{O}(n^2)$ of self-attention over long sequences, while the decoding stage is memory-intensive as it requires storing and repeatedly accessing the full KV cache.

Let $\mathbf{X}^v = \{\mathbf{x}_1^v, \ldots, \mathbf{x}_{|\mathbf{X}^v|}^v\}$ and $\mathbf{X}^t = \{\mathbf{x}_1^t, \ldots, \mathbf{x}_{|\mathbf{X}^t|}^t\}$ represent the video and text tokens, respectively, with video tokens preceding the text. For each transformer layer $l \in \{1, \ldots, L\}$, the KV cache comprises tensors $\mathbf{K}^{(l)}, \mathbf{V}^{(l)} \in \mathbb{R}^{(|\mathbf{X}^v|+|\mathbf{X}^t|) \times n_h \times d_h}$, where $n_h$ is the number of attention heads and $d_h$ is the per-head dimensionality. For example, let 8B InternVL-2.5 (Chen et al., 2025a) model process a one-hour video at 1 frame per second, the total required memory is around 400GB (see subsection D.2). This memory footprint makes KV cache storage a critical bottleneck in VideoLLM inference, significantly limiting the maximum processable video length and constraining the feasible batch size.

### 2.2 Long Video Processing

Multimedia container formats like MP4 or MKV bundle all the elements required for media playback, including video streams, audio streams, subtitles, and metadata (Koenen, 1999). In these containers, videos are stored as compressed bit streams (Koenen, 1999; Wiegand et al., 2003). In multimedia processing libraries like FFmpeg (Tomar, 2006), video decoding is described by a queue $\mathcal{D}$ that enqueues fixed-sized blocks of the bit stream, called *packets*, as input and dequeues video frames. We denote a bit stream $\mathcal{S} = (p_0, p_1, \ldots, p_{n-1})$ and a video $\mathcal{V} = (f_0, f_1, \ldots, f_{m-1})$ as ordered lists of packets and frames, respectively. Each frame $f_i$ is a tensor containing 8-bit integers of shape $(3 \times h \times w)$, where $h$ is the pixel height and $w$ is the pixel width. In general, *packets are not frame aligned*, enqueueing a single packet to the decoder can cause the decoder to output zero, one or potentially multiple frames (Wiegand et al., 2003). This is because frames require varying amounts of information to encode, and therefore cannot be aligned to fixed-sized packets. Furthermore, video frames are not encoded independently in bit stream, as surrounding frames contain redundant information. Therefore, the video encoder encodes the residual of the frame in the bitstream, instead of the frame itself (Wiegand et al., 2003; Sullivan et al., 2012). For this reason, video decoding is a largely sequential process, where

---

The encoded residual of a frame may require information from previous or future frames to decode (Wiegand et al., 2003).

previous frames must decoded first and then the residual information encoded in the bit stream can be used to decode the next frame (Wiegand et al., 2003). Although the video encoder may also reorder frames in the bit stream for efficiency, the decoder always outputs frames in the order that they should be displayed during playback (Tomar, 2006).

**Packet and Frame Metadata.** Although metadata is not directly encoded in the bit stream or frame itself, for simplicity, we denote metadata corresponding to packets or frames as if they are fields. The packet and frame metadata is stored in the container, not the bit stream (Koenen, 1999). The presentation timestamp (pts) of a frame is a 64-bit unsigned integer that represents when the frame should be displayed to a user (Tomar, 2006). Most formats do not include global frame positioning information in metadata. We instead use Equation (1) to rescale the presentation timestamp for a frame $f$ to obtain $f$'s index $i$ in $\mathcal{V}$.

$$i = \left\lfloor \frac{(m-1) \cdot f.\text{pts}}{pts_{max} - pts_{min}} \right\rceil \tag{1}$$

$pts_{max}$ and $pts_{min}$ are the minimum and maximum presentation timestamp for the video stream. Each packet has a *keyframe* flag that marks that video decoding can begin from its position (Koenen, 1999; Tomar, 2006).

### 2.3 Keyframes and Seeking

As video decoding relies on surrounding frames, it is a sequential process. However, during playback, users may want to navigate and skip through the video. To support this, the bit stream contains *keyframes*, which act as reset points from which video decoding can begin. Keyframes are encoded at semi-regular intervals in $\mathcal{S}$, usually a few seconds apart. To use keyframes to navigate in $\mathcal{S}$, we use the SEEK subroutine. SEEK($\mathcal{S}$, $pts$) finds the keyframe packet $p_i \in \mathcal{S}$ such that decoding from $p_i$ yields all $f$ such that $f.\text{pts} \geq pts$. However, seeking introduces overhead, as it requires flushing decoder buffers and reinitializing state (Tomar, 2006).

---

**Algorithm 1** Seek-based video decoding

---

**Require:** Bit stream $\mathcal{S}$, Ordered set $\mathcal{I}$, Video Decoder $\mathcal{D}$, $h$, $w$
1: Allocate memory block $\boldsymbol{F}$ of size $|\mathcal{I}| \times 3 \times h \times w$
2: **for** $i \in \mathcal{I}$ **do**
3:     Estimate $pts$ of $f_i$
4:     $p_i \leftarrow$ SEEK($\mathcal{S}$, $pts$)                                    ▷ Seek to the keyframe before $f_i$ in $\mathcal{S}$
5:     Decode $p_i, p_{i+1}, \ldots$ until $\mathcal{D}$ outputs $f_i$
6:     Write $f_i$ to $\boldsymbol{F}$
7: **return** $\boldsymbol{F}$

---

Algorithm 1 is a standard approach when decoding video for machine learning (Distributed (Deep) Machine Learning Community, 2019; PyTorch Team, 2025). For each desired frame $f_i$, given by selected indices in $\mathcal{I} \subseteq \{1, 2, \ldots, m-1\}$, the algorithm does the following: It seeks for the keyframe closest to $f_i$ in $\mathcal{S}$, and then it decodes packets until $\mathcal{D}$ outputs $f_i$. $f_i$ is saved in the buffer $\boldsymbol{F}$. This algorithm performs well for sparse access patterns, as if there are large gaps between desired frames, seeking before decoding each frame is ideal.

## 3 Method

In this section, we introduce QUICKVIDEO, which consists of three main components:

### 3.1 QuickCodec: Long Video Decoding for VideoLLMs

Given a bitstream $\mathcal{S}$ for a video $\mathcal{V} = (f_1, f_2, \ldots, f_m)$, where each frame has height $h$, width $w$, and the desired degree of concurrency is $c$, our goal is to compute $\boldsymbol{F}$ such that for all $j \in 0, 1, \ldots, |\mathcal{I}| - 1$, we have $\boldsymbol{F}_j = f_{\mathcal{I}[j]}$, where $\mathcal{I} \subseteq 1, 2, \ldots, m-1$. In other words, $\boldsymbol{F}$ is a packed tensor containing all the frames selected by $\mathcal{I}$. We assume $m$ is known from container metadata or can be estimated using $pts_{max}$ and $pts_{min}$.

The efficiency of our algorithm relies on two key observations:

(1) It is faster to use $c$ cores to decode $c$ short videos than to use $c$ cores to sequentially decode a single long video. Video decoding for human playback focuses on the latter case, as humans watch earlier frames while later frames are decoded. However, due to inter-frame dependencies, sequential video decoding is difficult to parallelize (Wiegand et al., 2003). In contrast, VideoLLMs require the entire video input to be loaded upfront. Therefore, we can decompose the loading of a long video $\mathcal{V}$ into loading $c$ short videos that collectively span $\mathcal{V}$. However, decoding cannot start at arbitrary frames—it must begin at keyframes. The KEYFRAME INTERVALS subroutine (Appendix A) parses the metadata of $\mathcal{S}$ and computes $c$ approximately equal-length intervals, starting and ending at keyframes, that cover the entire video. We parallelize over these intervals in Algorithm 2.

(2) VideoLLMs typically sample frames at a short, regular interval, usually 1–2 FPS (Bai et al., 2025), which is often smaller than the interval between keyframes in standard codecs. Consequently, seek-based decoding must still decode from all keyframes, leading to redundant seeks. Our algorithm requires only one seek operation per core, instead of a number of seeks proportional to the number of frames.

---

**Algorithm 2** QUICKCODEC

**Require:** Bit stream $\mathcal{S}$, Ordered set $\mathcal{I}$, Video Decoder $\mathcal{D}$, $h$, $w$, $c$, $m$

1: $\mathcal{J} \leftarrow$ KEYFRAME INTERVALS$(\mathcal{S}, c)$         ▷ $t$ intervals that start and end on a keyframe
2: Allocate shared memory $\boldsymbol{F}$ of size $|\mathcal{I}| \times 3 \times h \times w$
3: Initialize memory offset map $M$
4: **for** $k \in \{0, 1, \ldots, |\mathcal{I}| - 1\}$ **do**
5:      $M[\mathcal{I}[k]] \leftarrow k$         ▷ Maps frame index to memory offset in $\boldsymbol{F}$
6: **for all** $(pts_{start}, pts_{end}) \in \mathcal{J}$ **in parallel do**         ▷ Parallelize over $t$ intervals
7:      $p_i \leftarrow$ SEEK$(\mathcal{S}, pts_{start})$         ▷ Seek to the packet at the start of the keyframe interval
8:      **repeat**
9:          **while** $\mathcal{D}$ not empty **do**
10:              $f \leftarrow \mathcal{D}.\text{dequeue}()$
11:              **if** $f.\text{pts} \geq pts_{end}$ **then**
12:                  **break**
13:              Compute $i$ with equation 1
14:              **if** $i$ in $M$ **then**
15:                  $o \leftarrow M[i]$         ▷ Get the memory offset for $f_i$ in $\boldsymbol{F}$
16:                  $\boldsymbol{F}_o \leftarrow f$         ▷ Write frame into shared memory tensor
17:          $\mathcal{D}.\text{enqueue}(p_i)$
18:          $p_i \leftarrow p_{i+1}$         ▷ Get next packet in bit stream $\mathcal{S}$
19:      **until** $f.\text{pts} \geq pts_{end}$
20: **return** $\boldsymbol{F}$

---

Algorithm 2 presents the core of our video decoding algorithm. The algorithm begins by using metadata to compute $c$ keyframe-aligned intervals that span the video (line 1). Lines 2–5 initialize a shared memory block $\boldsymbol{F}$ and compute a dictionary $M$ that maps indices of selected frames to unique memory offsets in $\boldsymbol{F}$. We then decode the long video in $c$ parallel intervals (lines 6–19). Video decoding starts by seeking to the start of each interval $pts_{start}$, which is guaranteed to be a keyframe (line 7). Packets are enqueued for decoding (lines 17–18) until the decoder yields frames for processing (line 9). If the timestamp of a dequeued frame is greater than or equal to the interval endpoint $pts_{end}$, parallel processing ends (lines 11–12, 19). As the intervals in $\mathcal{J}$ span $\mathcal{S}$, $pts_{min}$ and $pts_{max}$ are given by the smallest and largest values in $\mathcal{J}$, respectively. We use Equation (1) to compute the index $i$ of $f$ (line 13). Finally, we save $f$ to $\boldsymbol{F}$ if $f$ is a selected frame (lines 14–16). Because decoding from a keyframe yields all frames with greater $pts$ values, and $\mathcal{D}$ outputs frames in $pts$ order, when the parallelized loop exits (line 19), all selected frames with $pts$ in the interval $[pts_{start}, pts_{end})$ will have been output by $\mathcal{D}$ and saved to $\boldsymbol{F}$. Thus, as $\mathcal{J}$ spans $\mathcal{S}$, when the algorithm returns, $\boldsymbol{F}$ will contain all selected frames.

### 3.2 QuickPrefill: Efficient Group-Based Prefilling for VideoLLMs

After decoding the video bitstream into packed tensors, they are fed into the VideoLLM for inference. However, LLM generation with long contexts is a well-known challenge due to high memory usage and computational cost. To address this, we introduce QUICKPREFILL, a grouped prefilling and KV cache pruning method that accelerates processing and significantly reduces memory requirements.

**Group-Based Prefill** Let $\mathbf{X}^v = \mathbf{x}_1^v, \ldots, \mathbf{x}_{|\mathbf{X}^v|}^v$ denote the sequence of video tokens, where $|\mathbf{X}^v|$ is the total number of tokens, and each token $\mathbf{x}_i^v \in \mathbb{R}^d$ is a $d$-dimensional vector. To reduce memory overhead during prefilling, we partition the video token sequence into $G$ disjoint groups: $\mathbf{X}^v = \mathbf{X}_1^v, \ldots, \mathbf{X}_G^v$, where each group $\mathbf{X}_g^v$ contains approximately $N_g = \frac{|\mathbf{X}^v|}{G}$ tokens. Instead of processing the entire sequence at once, we sequentially prefill each group and store the corresponding key-value (KV) cache as $\mathbf{K}_g^{(l)}, \mathbf{V}_g^{(l)}$ for each transformer layer $l$. This strategy significantly reduces peak activation memory usage by a factor of $G$ and remains effective even when combined with efficient attention mechanisms such as FlashAttention. Empirically, it enables hour-long video understanding while keeping GPU memory usage within practical limits (e.g., reducing memory overhead by over 100 GB; see subsection D.2).

**Group-Based KV Cache Pruning** While group-based prefill reduces peak activation memory, the KV cache memory remains a major bottleneck. To address this, we prune unimportant KV cache vectors when processing each group, maintaining a retention ratio $\rho \in (0, 1]$. This reduces KV cache memory usage by a factor of $\frac{1}{\rho}$.

The pruning decision is based on an importance score function $\mathbf{s}$, which produces an ordered list of KV entries. We select the top-$k$ KV entries until the retention ratio is satisfied:

$$\tilde{\mathbf{K}}_g^{(l)} = \mathbf{K}_g^{(l)}[I_g^{(l)}], \quad \tilde{\mathbf{V}}_g^{(l)} = \mathbf{V}_g^{(l)}[I_g^{(l)}], \quad \text{where } I_g^{(l)} = \text{TopK}\left(\mathbf{s}(\mathbf{K}_g^{(l)}, \mathbf{V}_g^{(l)}), ; k = \rho \cdot N_g\right) \tag{2}$$

where TopK returns the indices of the top-$k$ entries. We consider several heuristic importance functions $\mathbf{s}$ from prior works (Devoto et al., 2024; Guo et al., 2024; Zhang et al., 2023). In this paper, we primarily use the following three: 1) Key Norms (small): $\mathbf{s} = -L_2(\mathbf{K}_g^{(l)})$; 2) Value Norms: $\mathbf{s} = L_2(\mathbf{V}_g^{(l)})$; 3) Attention Scores: $\mathbf{s} = \text{matmul}(\mathbf{K}_g^{(l)}, \mathbf{Q}^{(l)})$. Here, $L_2$ denotes the L2-norm function, and $\mathbf{Q}^{(l)} \in \mathbb{R}^{|\mathbf{X}^t| \times (n_h \times d_h)}$ is the query vector of text tokens in layer $l$. We adopt Key Norms (small) as the default importance function in QUICKPREFILL due to its strong performance. Using key norms also allows us to use efficient fused attention implementations, such as FlashAttention (Dao, 2023).

### 3.3 Overlapping QuickCodec and QuickPrefill

The preceding sections introduced two complementary components: QUICKCODEC for CPU-based video decoding and QUICKPREFILL for GPU-based group-wise prefilling. However, running these components sequentially underutilizes resources—GPUs remain idle during video decoding, and CPUs are underutilized during prefilling. To address this inefficiency, we propose an overlapped execution scheme that enables concurrent processing across CPU and GPU resources.

To achieve this, we slightly adapt frame loading: Instead of using $c$ cores to load $c$ intervals, we divide $\mathcal{V}$ into $s$ intervals, where $s \gg c$, using KEYFRAME INTERVALS$(\mathcal{V}, s)$. We then load the frames from the $s$ intervals using $c$ cores, prioritizing earlier intervals so that frames corresponding to the first blocks of video are available sooner. This allows us to exploit QUICKCODEC's fast video decoding while ensuring that early frames in $\mathcal{V}$ are prioritized for QUICKPREFILL. Once the frames required for the first group are loaded, QUICKPREFILL begins processing immediately, while QUICKCODEC continues decoding subsequent frames in the background. After QUICKPREFILL finishes processing a group, it stores the resulting KV cache and checks whether QUICKCODEC has loaded the frames required for the next group. If so, QUICKPREFILL starts processing the next group immediately. This design forms a producer-consumer pipeline between CPU decoding and GPU prefilling, ensuring the GPU is only idle if it is waiting for the CPU to finish decoding the next set of frames.

The performance improvement of this overlap strategy can be formalized. Let $t_{dec}^i$ and $t_{prefill}^i$ denote the time to decode and prefill a group of frames, respectively. With our overlap strategy, the execution time is modeled by Equation (3).

$$t_{total} = \sum_i^G \max(t_{dec}^i, t_{prefill}^i) + \Delta \tag{3}$$

$\Delta$ is a small latency introduced by QUICKCODEC's metadata parsing and startup. Since each group contains a small number of frames and tokens, this strategy achieves near-optimal overlap between CPU and GPU resources, resulting in substantial speedup for hour-long video processing. Note that some VideoLLMs include additional preprocessing steps (e.g., position embedding calculation or normalization), which we do not include in this analysis (Bai et al., 2025).

## 4 Experiments

We evaluate QuickVideo's performance on practical long video understanding tasks. In section 4.1, we benchmark QUICKCODEC against existing frameworks. We also examine the limitations of QUICKCODEC, identifying use cases where seek-based frameworks (Algorithm 1) outperform our method. Next, in section 4.2, we evaluate the performance of QUICKPREFILL across four long video understanding benchmarks, analyzing the trade-off between accuracy and efficiency. Finally, in section 4.3, we demonstrate that the prefill and video decoding stages can be almost entirely overlapped, effectively reducing end-to-end inference time by nearly a minute for long video inputs.

### 4.1 QuickCodec Results

**Video Loading Speed.** We benchmark the time required to load an hour-long 24 FPS 1920x1080p HD video, sampled at 1 FPS and resized to 448x448 pixels. The video is a one-hour segment of a popular movie encoded with default FFmpeg settings using H.264, the most widely used codec (Kerdranvat et al., 2020). Sampling frames at 1–2 FPS is a standard practice in VideoLLMs, balancing computational efficiency with task performance (Bai et al., 2025). We resize frames to 448x448 pixels, which matches the maximum per-frame resolution used in most VideoLLMs (Zhu et al., 2025a; Chen et al., 2025a). All experiments are

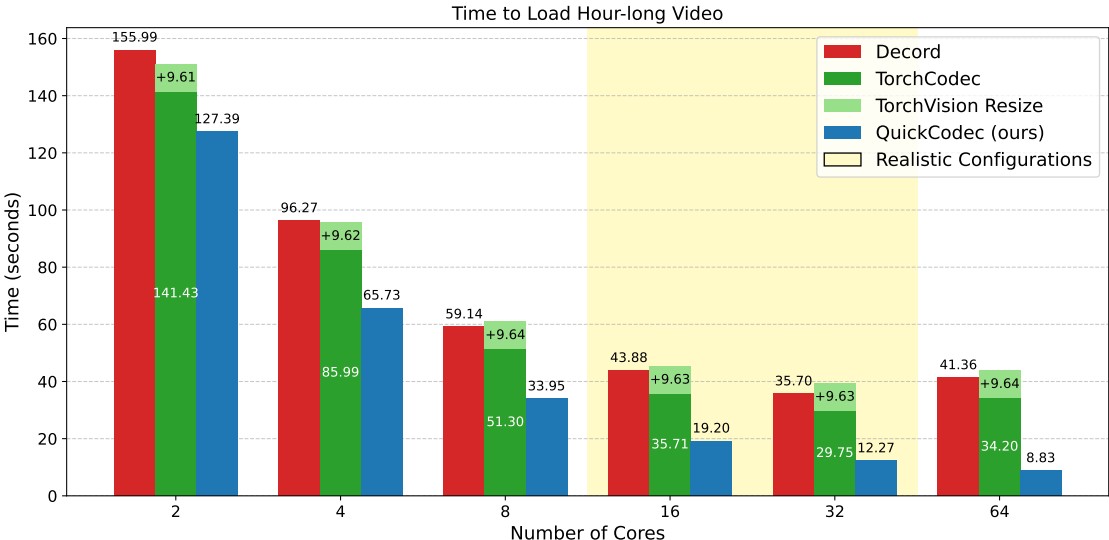

Figure 2: Speed comparison of Decord, TorchCodec (with Resize), and QUICKCODEC when loading hour-long videos. We ablate across different levels of parallelization (core counts).

conducted on an AWS m7a.16xlarge instance. Each timing result is averaged over five runs, with a 95% confidence interval no greater than 0.5 seconds.

We compare QuickVideo against two widely used video decoding frameworks:

**Decord** (Distributed (Deep) Machine Learning Community, 2019): A multimedia loading framework designed for machine learning applications. While no longer actively maintained, Decord remains integrated into popular libraries like Hugging Face's *Transformers* (Wolf et al., 2020) and, by extension, inference frameworks such as *vLLM* (Kwon et al., 2023).

**TorchCodec** (PyTorch Team, 2025): A work-in-progress library from the PyTorch team designed to offer faster multimedia processing than TorchVision (maintainers & contributors, 2016). TorchCodec lacks some features of mature frameworks, such as built-in support for frame resizing. Thus, we report timings that combine TorchCodec loading with a resizing step via TorchVision. TorchCodec is not optimized for decoding with more than 16 cores; we observe that increasing core count beyond 16 can even degrade performance. In collaboration with the TorchCodec team, algorithm 2 has been implemented into TorchCodec, we do not use this setting in our timings.

As shown in Figure 2, QUICKCODEC outperforms other libraries across varying core counts. While other frameworks plateau at 16 cores, QUICKCODEC scales up to 64 cores. We highlight the 16- and 32-core cases as these are the most common configurations in practical deployments; most compute providers allocate between 16 and 32 CPU cores per GPU (Google, 2025; Amazon, 2025; Microsoft, 2025). At 16–32 cores, QUICKCODEC is 2–3× faster than other libraries when loading an hour-long video, reducing video loading time by over 20 seconds.

**Speed Across Video Durations.** Our framework relies on pre-computing intervals and sufficient keyframes for parallelization. Therefore, we expect reduced benefits for shorter videos. We benchmark QUICKCODEC on videos of varying lengths, from 1 minute to 1 hour, using the same source video (an hour-long movie) cut to different durations. All tests use 1 FPS sampling and 16 cores for video decoding. Results are averaged over five runs on an AWS m7a.16xlarge instance, with a 95% confidence interval of at most 0.2 seconds. We find that QUICK-CODEC is consistently faster than other frameworks for videos longer than 1 minute (Figure 3). Its advantage grows with video length—QUICKCODEC is 1.7× faster than Decord for a 10-minute video and 2.1× faster for a 1-hour video. We further discuss scenarios where seek-based decoders outperform QUICKCODEC in Appendix B.

Figure 3: Video decoding performance across different video durations (1 FPS sampling).

### 4.2 QuickPrefill Results

We evaluate QUICKPREFILL on four long video understanding benchmarks, with videos ranging from minutes to hours: VideoMME (Fu et al., 2024), LongVideoBench (Wu et al., 2024), LVBench (Wang et al., 2024), and MLVU (Zhou et al., 2024a). All generations use greedy sampling, and results are reported via the `lmms-eval` framework (Zhang et al., 2024). Experiments are run on the Qwen2.5-VL-7B-Instruct model (Bai et al., 2025) using a single A100 (40GB) GPU with 8 replicas.

**Effectiveness of Different KV Cache Pruning Methods.** We evaluate the impact of various KV cache pruning strategies on model accuracy, as summarized in Table 1. We compare several pruning techniques against a no-pruning baseline ($\rho = 1$), fixing the retention ratio $\rho$ at 0.5 and the group size at 16 frames. The *Key Norms (small)* method achieves the best balance between efficiency and accuracy, retaining over 95% of the model's original performance while halving the KV cache size and computation. In the 1024-frame setting, it retains over 98% of the original performance. Notably, this method outperforms query-attention-based token selection strategies. While prior work (Devoto et al., 2024) has shown that negative L2 norms of keys

Table 1: Effectiveness of different KV cache pruning methods in the group-based prefilling scenario. We use the *Key Norms (small)* as the default KV cache pruning method for QUICKPREFILL due to its superior performance and query-agonistic nature.

| Group Size #Frames | KV Pruning | $\rho$ | VideoMME w/o subtitle | LongVideoBench val | LVBench test | MLVU dev | Avg | Performance |
|---|---|---|---|---|---|---|---|---|
| 64 Frames | | | | | | | | |
| - | - | 1 | 62.41 | 59.69 | 40.09 | 63.86 | 56.51 | 100.00% |
| 16 | Value Norms | 0.5 | 47.63 | 35.98 | 30.92 | 31.38 | 36.48 | 64.55% |
| 16 | Attention Scores | 0.5 | 58.63 | 52.95 | 37.83 | 59.87 | 52.32 | 92.58% |
| 16 | *Key Norms (small)* | 0.5 | 60.56 | 56.17 | 37.70 | 62.34 | 54.19 | **95.90%** |
| 128 Frames | | | | | | | | |
| - | - | 1 | 66.41 | 60.96 | 42.87 | 66.86 | 59.27 | 100.00% |
| 16 | Value Norms | 0.5 | 48.56 | 37.32 | 30.73 | 38.51 | 38.78 | 65.42% |
| 16 | Attention Scores | 0.5 | 60.96 | 55.20 | 39.70 | 64.36 | 55.06 | 92.89% |
| 16 | *Key Norms (small)* | 0.5 | 63.41 | 58.19 | 39.57 | 64.99 | 56.54 | **95.39%** |
| 256 Frames | | | | | | | | |
| - | - | 1 | 65.78 | 61.56 | 43.90 | 68.65 | 59.97 | 100.00% |
| 16 | Value Norms | 0.5 | 48.33 | 38.89 | 31.38 | 37.74 | 39.08 | 65.17% |
| 16 | Attention Scores | 0.5 | 62.52 | 57.22 | 41.96 | 67.27 | 57.24 | 95.45% |
| 16 | *Key Norms (small)* | 0.5 | 64.04 | 60.21 | 41.90 | 66.73 | 58.22 | **97.08%** |
| 1024 Frames | | | | | | | | |
| - | - | 1 | 62.00 | 60.43 | 42.29 | 63.48 | 57.05 | 100.00% |
| 16 | Value Norms | 0.5 | 47.37 | 33.66 | 29.18 | 32.65 | 35.71 | 62.60% |
| 16 | Attention Scores | 0.5 | 62.22 | 58.49 | 42.03 | 64.45 | 56.80 | **99.56%** |
| 16 | *Key Norms (small)* | 0.5 | 59.99 | 61.59 | 40.80 | 64.76 | 56.78 | 99.53% |

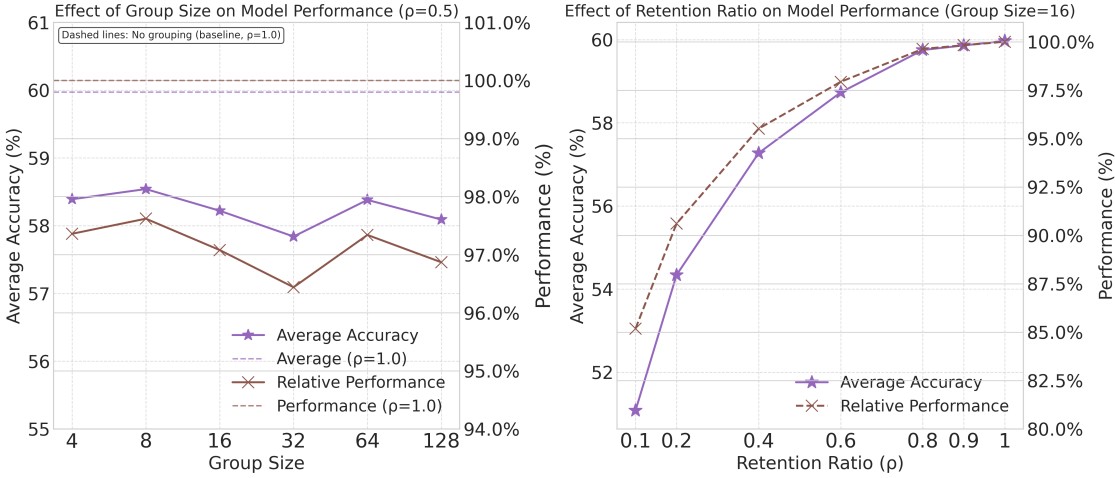

Figure 4: Ablation study on group size and retention ratio. Data from Table 2.

correlate strongly with attention scores in text-only LLMs, our results extend this finding to VideoLLM prefilling, highlighting the generalizability and practical utility of key norm-based pruning.

We also conduct ablation studies on group size and retention ratio $\rho$ (see Appendix E). As shown in Table 2 and Figure 4, group size has minimal impact on model performance, while increasing $\rho$ consistently improves accuracy, approaching the no-pruning baseline. Smaller group sizes reduce activation memory, while lower $\rho$

Figure 5: Latency breakdown for video loading, prefill, and LLM decoding in end-to-end inference. We compare a baseline Qwen2.5-VL Bai et al. (2025) implementation, the same model with QUICKPREFILL and QUICKCODEC, and our block-overlapped design.

values reduce KV cache memory. These findings provide practical guidance for balancing memory efficiency and model accuracy based on system constraints.

### 4.3 Latency in End-to-End QuickVideo Inference

We integrate QUICKCODEC and QUICKPREFILL into a Qwen2.5-VL-7B-Instruct (Bai et al., 2025) inference pipeline. We evaluate two configurations: (1) loading the entire video with QUICKCODEC followed by QUICKPREFILL, and (2) our group-overlapped design. Latency for video loading, prefill, and LLM decoding is benchmarked in an end-to-end pipeline. For QUICKPREFILL, we use the Key Norms (small) pruning method with $\rho$ = 0.2 and set the group size to 32 frames. We use a 30-minute video (sampled at 1 FPS) as the baseline implementation runs out of memory with longer videos. Experiments are conducted on an H100 SXM GPU and AMC Ryzen 9 7950x CPU, allocating 16 cores for video processing. For the overlapped implementation, we use 64 intervals ($s$ in section 3.3) for parallelized loading. All timings are averaged over 5 runs. Figure 5 presents latency breakdowns for all three implementations. After applying QUICKCODEC, we significantly reduce video loading time from 25.0 seconds to 15.4 seconds. By overlapping video loading and prefill, we reduce latency by 39.45% when compared to the baseline. We find that once prefill and video decoding are overlapped, the combined stage becomes completely video-decoding bound.

## 5 Discussion and Related Work

**GPU support for video decoding.** Video decoding can be accelerated by GPU computing. However, due to interframe dependencies, the speedup is not nearly as large as GPU acceleration for AI computations PyTorch Team (2025). Furthermore, especially in the case of long video, GPU-based video decoding can result in device memory problems; the hour-long video we use for benchmarking (Section 4.1) is 3600×3×1920×800×1 byte ≈ 16.6 GB before being resized. This results in a significant portion of GPU resources being allocated to video tensors, and can cause CUDA out-of-memory errors if not handled delicately. For simplicity, most existing inference libraries default to using CPU for video decoding (Kwon et al., 2023; Wolf et al., 2020). More sophisticated pipelines, such as NVIDIA's Cosmos training, use dedicated hardware for handling the video processing (NVIDIA et al., 2025).

**Efficient VideoLLMs Inference.** Recent VideoLLMs (Lin et al., 2023; Li et al., 2024; Chen et al., 2025a) have demonstrated strong video understanding capabilities. Early models like Video-LLaVA (Lin et al., 2023) and VideoLLama-2 (Cheng et al., 2024) were limited to around 32 input frames due to constrained training data and unoptimized architectures. More advanced models such as Qwen2.5-VL (Bai et al., 2025) and InternVideo2.5 (Wang et al., 2025) can now handle hundreds of frames by adopting architectural innovations including Group Query Attention (GQA) (Ainslie et al., 2023), MRoPE (Bai et al., 2025), and Special Token Merging (Chen et al., 2025a), which reduce KV cache size and enhance temporal reasoning. Nonetheless, the KV cache and activation memory still grow linearly with context length, creating bottlenecks in hour-long video inference. Meanwhile, existing token pruning techniques either address only image-level contexts (Wen et al., 2025; Chen et al., 2024; Shang et al., 2024; Xing et al., 2024), or optimize for short prefill and long decoding scenarios (Devoto et al., 2024; Zhang et al., 2023; Xiao et al., 2023). In contrast, we target efficient prefill for millions of video tokens, introducing a method that achieves substantial memory savings and speedup

with minimal accuracy loss, thereby enabling scalable long video understanding on resource-constrained hardware.

## 6 Conclusion

We introduced **QuickVideo**, a framework to accelerate long video understanding. Our framework has three core contributions: **QuickCodec**: A systems framework for fast video loading, designed for VideoLLM frame sampling. **QuickPrefill**: An efficient algorithm for prefilling video tokens. **Co-design:** Lastly, we show that our video loading and prefill algorithm can be almost entirely overlapped, drastically reducing the time latency of these stages during inference. Overall, **QuickVideo** reduces time to infer a long video input by more than 3×. Our work advances the capabilities for real-time video understanding applications, addressing key efficiency challenges in long video inference.

**Broader Impact Statement**

As video has become the dominant modality of data, efficient long video understanding has extremely broad implications, both positive and negative. On the positive side, better long video understanding allows us to better interpret our digital landscape. In 2022, 30,000 hours of video were uploaded to YouTube every hour (Ceci, 2024). That number is absolutely much higher today. Without efficient long video understanding systems, we cannot understand our own digital artifacts, due to the scale at which we create them. Furthermore, long video understanding also has extremely compelling use-cases for information accessibility. A video-first internet is difficult to navigate for visually impaired people, with important information potentially only accessible in video format (Liu et al., 2021). Efficient, robust long video understanding presents can serve as a backbone for tools for assisting video understanding for the visually impaired. However, efficient long video understanding also has potentially negative effects. As people's lives are increasingly documented as video and uploaded to the internet, long video understanding models could become a tool for privacy intrusion (Feldstein, 2022).

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

## A  Parallelized Interval Algorithm

**Additional video decoding background.** The container contains various metadata about packets that we use during our interval parsing algorithm. For locality purposes, modalities such as audio and video are often *interleaved* in the bit stream $\mathcal{S}$. Therefore, it is important to filter out audio packets when parsing the metadata stream. As packets are not frame-aligned, the pts field does not exactly represent the display time of frame. Also, as packets can be reordered by the decoder, the first or last packets may not correspond to the first and last frames.

---

**Algorithm 3** Calculate Parallelized Intervals

---

1: **procedure** KEYFRAME INTERVALS($\mathcal{S}, c$)
2:     $\mathcal{K}, pts_{min}, pts_{max} \leftarrow$ SCAN PACKETS($\mathcal{S}$)          ▷ Scan packet metadata.
3:     $\mathcal{J} \leftarrow \{pts_{min}, pts_{max}\}$          ▷ Ordered list of keyframe intervals.
4:     $p \leftarrow \frac{1}{c}(pts_{max} - pts_{min})$          ▷ Evenly spaced intervals in the video.
5:     **for** $i \in 1, \ldots, c-1$ **do**
6:         $pts_{estimate} \leftarrow (c \times p) + pts_{min}$
7:         $j \leftarrow$ FINDINSERTIONINDEX($\mathcal{K}, pts_{estimate}$)
8:         **if** $|\mathcal{K}_{j-1} - pts_{estimate}| < |\mathcal{K}_j - pts_{estimate}|$ **then**
9:             $\mathcal{J} = \mathcal{J} \cup \{\mathcal{K}_{j-1}\}$
10:        **else**
11:            $\mathcal{J} = \mathcal{J} \cup \{\mathcal{K}_j\}$
       **return** $\mathcal{J}$
12: **procedure** SCAN PACKETS($\mathcal{S}$)          ▷ Scan bit stream to get timestamps.
13:     $pts_{min} \leftarrow -1$
14:     $pts_{max} \leftarrow \infty$
15:     $\mathcal{K} \leftarrow \varnothing$          ▷ Sorted set of keyframe timestamps.
16:     **for** $p_i \in \mathcal{S}$ **do**
17:         **if** $p_i$.type $\neq$ "video" **then**          ▷ Skip packets are not used to decode video.
18:             **continue**
19:         **if** $p_i$.pts = NULL **then**          ▷ Skip packets do not have pts metadata.
20:             **continue**
21:         **if** $p_i$.pts $< pts_{min}$ **then**
22:             $pts_{min} \leftarrow p_i$.pts
23:         **if** $p_i$.pts $> pts_{max}$ **then**
24:             $pts_{max} \leftarrow p_i$.pts
25:         **if** $p_i$.keyframe = True **then**
26:             $\mathcal{K} \leftarrow \mathcal{K} \cup \{p_i.$pts$\}$
27:     **return** $\mathcal{K}, pts_{min}, pts_{max}$

---

Algorithm 3 computes $c$ intervals that we can parallelize video decoding over. For effective parallelization, it is essential that these intervals are roughly length and keyframe-aligned, such that Algorithm 2 can seek to the start of each interval. SCAN PACKETS parses the metadata of the packet stream to find the location of all keyframes in $\mathcal{S}$, as well as the minimum and maximum pts in $\mathcal{S}$. If the packet does not belong to the video stream or the timestamp is NULL, the packet is skipped.

After finding the locations of keyframes on line 2, KEYFRAME INTERVALS computes $c$ intervals as follows: We calculate the length of $\frac{1}{c}$ of the stream, in pts units (line 4). On lines 5-10, we search for the keyframes closest to being $\frac{i}{c}th$ through the video, given by $pts_{estimate}$. FINDINSERTIONINDEX uses binary search to find where in the list of keyframes $pts_{estimate}$ would be inserted. After finding the insertion point $j$, the algorithm checks whether the keyframe before or after $j$ is closer to $pts_{estimate}$. The closest keyframe location is added to $\mathcal{J}$, the list of intervals. $\mathcal{J}[0] = pts_{min}$ and $\mathcal{J}[c-1] = pts_{max}$, to ensure that the intervals span the video. $\mathcal{J}[1], \mathcal{J}[2], \ldots \mathcal{J}[c-2]$ are keyframe-aligned and equally spaced. Therefore, $\mathcal{J}$, a list containing $c+1$ values, can be interpreted as $c$ intervals: $\mathcal{J}' = \{(\mathcal{J}[i], \mathcal{J}[i+1]) \mid i \in 0, 1, \ldots, c\}$.

## B    Effect of sampling rates on QuickCodec's efficiency

As QUICKCODEC does not seek between loading frames, all video frames are decoded during video loading. Conversely, seek-based frameworks skip decoding segments of video if there are large gaps between sampled frames. In Figure 6, we find that our framework has faster video loading when there is a 4 second or less gap between sampled frames. Our library performs best when using VideoLLM sampling rates (1-2 FPS). Currently, our implementation always loads the whole video, and therefore does not benefit significantly from sparse sampling patterns. Our implementation could be adapted to leverage seeking when it detects that the user has sampled with a large gap between frames, closing the performance gap with seek-based libraries PyTorch Team (2025); Distributed (Deep) Machine Learning Community (2019). This would make our library more flexible, and eliminate a potential performance sharp edge, where users accidentally use our QUICKCODEC for sparse sampling. We leave this as a direction for future library improvements.

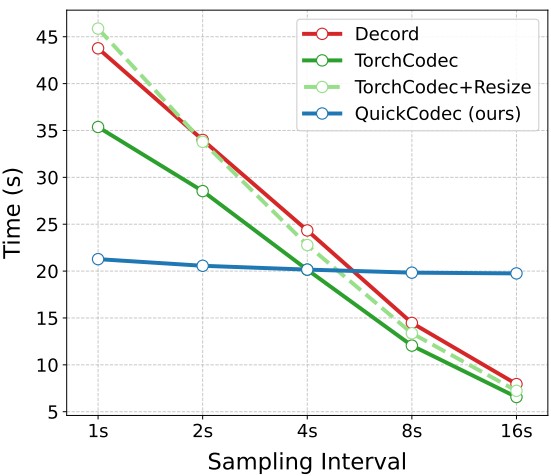

Figure 6: Video decoding performance for different video durations with 1 FPS sampling.

## C    Containers and Video Decoding

A multimedia container file format, like MP4 or MKV, bundles together all the elements required for media playback, including video streams, audio streams, subtitles, images, and metadata (Koenen, 1999). Video streams are compressed into bit streams by *codecs* . The bit streams are formatted in standards like H.264 (Wiegand et al., 2003) and H.265 (Sullivan et al., 2012). A codec consists of two algorithms: a video encoding algorithm that takes in a sequence of frames and outputs a compressed bit stream and a video decoding algorithm that takes the bit stream as input and outputs video frames. We focus video decoding, as it is the required operation before the video can be used as a VideoLLM input.

## D    QuickPrefill Efficiency Analysis Details

### D.1    Activation Memory Analysis

The activation memory of modern LLM architecture mainly comes from two components of each transformer block: **1)** Attention Block and **2)** MLP Block. We analyze the potential activation memory usage in formulas in the followings and show that group-based prefilling can effectively reduce the activation memory by $G$ times, where $G$ is the number of groups.

**Attention Block**   Modern LLMs commonly adopt FlashAttention (Dao, 2023), a memory-efficient attention algorithm that computes exact attention with reduced memory usage by fusing multiple steps and processing attention in blocks. While the naive attention implementation would instantiate the full attention matrix $A \in \mathbb{R}^{S \times S}$, FlashAttention avoids this by computing attention block by block. Let $Q, K, V \in \mathbb{R}^{B \times S \times d_{\text{head}}}$ denote the query, key, and value tensors respectively, with $n_h$ heads and $d_{\text{head}} = \frac{d_{\text{model}}}{n_h}$. FlashAttention divides the input sequence into blocks of size $B_c$ (for keys/values) and $B_r$ (for queries) to process attention efficiently within GPU memory constraints. Following (Dao, 2023), the dominant activation memory in FlashAttention comes from storing $Q, K, V$. The block-based processing means that at any given time, only blocks of the attention matrix of size $B_r \times B_c$ are materialized in memory. Assume using `float16` data type, the total activation memory can be expressed as:

$$\mathcal{M}_{\text{attn}} \approx (3B \cdot S \cdot n_h \cdot d_{\text{head}} + B \cdot n_h \cdot B_r \cdot B_c) \cdot 2 \text{ bytes} \tag{4}$$

The first term accounts for storing $\boldsymbol{Q}$, $\boldsymbol{K}$, and $\boldsymbol{V}$ tensors, while the second term accounts for the block of attention matrix being processed. With appropriate block sizes $B_r$ and $B_c$ (typically set based on GPU memory constraints), the second term remains relatively small. Assuming $B = 1$, $S = (|\mathbf{X}^v| + |\mathbf{X}^t|) \approx 921600$, $d_{\text{model}} = 4096$, $n_h = 8$, $B_r = B_c = 1024$, we compute:

$$\mathcal{M}_{\text{attn}} = (3 \cdot 1 \cdot 921600 \cdot 8 \cdot 512 + 1 \cdot 8 \cdot 1024 \cdot 1024) \cdot 2 \text{ bytes} \tag{5}$$

$$= 60,584,722,432 \text{ bytes} \tag{6}$$

$$\approx \boxed{21.1 \text{ GB}} \tag{7}$$

While FlashAttention significantly reduces memory requirements compared to naive attention implementation, this analysis shows it still consumes substantial memory for very long sequences. With group-based prefilling using $G = 225$ groups, we can reduce the sequence length $S$ by $G$ times, reducing $\mathcal{M}_{\text{attn}}$ from 21.1 GB to approximately **0.09** GB. This dramatic reduction enables the processing of extremely long sequences that would otherwise be infeasible.

**MLP Block**   The SwiGLU (Swish-Gated Linear Unit) (Shazeer, 2020) enhances transformer models through improved gating mechanisms and has been adopted as the default MLP architecture in many popular LLMs including InternVL2.5 and Qwen2.5 series (Bai et al., 2025; Chen et al., 2025a). For input representation $\boldsymbol{x} \in \mathbb{R}^{d_{\text{model}}}$, the SwiGLU operation is defined as:

$$\text{SwiGLU}(\boldsymbol{x}) = \boldsymbol{W}_{\text{down}}(\text{SiLU}(\boldsymbol{W}_{\text{gate}}\boldsymbol{x}) \odot \boldsymbol{W}_{\text{up}}\boldsymbol{x}) \tag{8}$$

where $\boldsymbol{W}_{\text{gate}}, \boldsymbol{W}_{\text{up}} \in \mathbb{R}^{d_{\text{ff}} \times d_{\text{model}}}$, $\boldsymbol{W}_{\text{down}} \in \mathbb{R}^{d_{\text{model}} \times d_{\text{ff}}}$, and $\text{SiLU}(x) = x \cdot \sigma(x)$ with $\sigma(x) = \frac{1}{1+e^{-x}}$.

For a batch of sequences, activation memory analysis reveals requirements at each computational step. With batch size $B$, sequence length $S$, hidden dimension $d_{\text{model}}$, intermediate dimension $d_{\text{ff}}$, and data type `float16`, the total activation memory for a single SwiGLU layer is:

$$\mathcal{M}_{\text{act}} = (B \cdot S \cdot (2d_{\text{model}} + 4d_{\text{ff}})) \cdot 2 \text{ bytes} \tag{9}$$

For a one hour video sampled with 1 FPS (3600 frames in total), parameters can be set $B = 1$, $S = (|\mathbf{X}^v| + |\mathbf{X}^t|) \approx 921600$, $d_{\text{model}} = 4096$, and $d_{\text{ff}} = 14336$:

$$\mathcal{M}_{\text{act}} = (1 \cdot 921600 \cdot (2 \cdot 4096 + 4 \cdot 14336)) \cdot 2 \text{ bytes} \tag{10}$$

$$= 241,591,910,400 \text{ bytes} \tag{11}$$

$$\approx \boxed{112.5 \text{ GB}} \tag{12}$$

This substantial memory requirement highlights the computational challenges in deploying SwiGLU-based models for high-resolution inputs with extended sequence lengths. However, if we prefill the tokens group by group, we can reduce the $S$ by $G$ times, and thus reduce the activation memory $\mathcal{M}_{\text{act}}$ by $G$ times. Assuming each group contains tokens of 16 frames, then $G = \frac{3600}{16} = 225$ and we can reduce $\mathcal{M}_{\text{act}}$ from 112.5 GB to **0.5** GB, which is a substantial improvement.

## D.2   KV cache Memory Analysis

When using InternVL2.5-8B (Chen et al., 2025a), with each frame encoded as 256 tokens ($|V| = 3,600 \times 256 = 921,600$), and $|Q| = 256$ text tokens, $L = 28$ layers, $n_h = 8$ heads, and $d_h = 512$, the total memory required to store the KV cache in `float16` precision is:

$$\text{Memory} = 2 \times L \times (|\mathbf{X}^v| + |\mathbf{X}^t|) \times n_h \times d_h \times 2 \text{ bytes} \approx \boxed{393.9 \text{ GB}}. \tag{13}$$

# E Ablation Study on Group Size and Retention Ratio

Table 2: Ablation study of different group sizes and retention ratio $\rho$. We use *Key Norms (small)* as the KV pruning method here.

| Group Size | $\rho$ | VideoMME | LongVideoBench (val) | LVBench | MLVU (dev) | Avg | Performance |
|---|---|---|---|---|---|---|---|
| | | | Varying Group Size | | | | |
| - | 1 | 65.78 | 61.56 | 43.90 | 68.65 | 59.97 | 100.00% |
| 4 | 0.5 | 63.78 | 60.36 | 42.61 | 66.81 | 58.39 | 97.36% |
| 8 | 0.5 | 64.00 | 60.88 | 42.35 | 66.94 | 58.54 | 97.62% |
| 16 | 0.5 | 64.04 | 60.21 | 41.90 | 66.73 | 58.22 | 97.08% |
| 32 | 0.5 | 63.59 | 59.46 | 41.51 | 66.78 | 57.84 | 96.44% |
| 64 | 0.5 | 63.89 | 60.51 | 42.29 | 66.83 | 58.38 | 97.34% |
| 128 | 0.5 | 63.56 | 59.24 | 42.61 | 66.97 | 58.09 | 96.87% |
| | | | Varying Retention Ratio $\rho$ | | | | |
| 16 | 1 | 65.78 | 61.56 | 43.90 | 68.65 | 59.97 | 100.00% |
| 16 | 0.1 | 55.89 | 53.40 | 36.02 | 59.02 | 51.08 | 85.18% |
| 16 | 0.2 | 59.74 | 56.47 | 39.57 | 61.58 | 54.34 | 90.61% |
| 16 | 0.4 | 63.22 | 58.94 | 41.19 | 65.75 | 57.27 | 95.51% |
| 16 | 0.6 | 64.74 | 60.81 | 41.90 | 67.48 | 58.73 | 97.93% |
| 16 | 0.8 | 65.70 | 61.41 | 43.51 | 68.37 | 59.75 | 99.63% |
| 16 | 0.9 | 65.85 | 61.18 | 43.71 | 68.70 | 59.86 | 99.82% |

# F HourVideo Results

HourVideo (Chandrasegaran et al., 2024) is a video-understanding benchmark designed to test the reasoning, perception, summarization and navigation abilities of VideoLLMs. In Table 3, we compare our best performing pruning method, *Key Norms (small)*, to a no pruning baseline. We use 256 frames and group size of 16 frames, as we found that is the strongest setting for the baseline (Table 1). The overall score is computed by weighting each category by its respective number of samples. We note that the navigation category requires answering with an image, which our model does not support.

# G QuickVideo CPU and GPU Memory Profiles

We provide a CPU memory profile corresponding to Figure 5. QuickVideo's CPU memory is made up of 3 components: the torch main process, QuickCodec workers, and the shared memory block into which video frames are loaded (line 2 in Algorithm 2). The torch main process has consistent memory utilization (RSS

| Method | Reasoning | Perception | Summarization | Navigation | Overall |
|---|---|---|---|---|---|
| No Pruning | 31.76 | 32.38 | 30.49 | 20.00 | 31.47 |
| *Key Norms (small)* | 33.09 | 28.98 | 31.71 | 17.50 | 31.13 |
| Performance | 104.2% | 89.5% | 104.0% | 87.5% | 98.9% |

Table 3: Results on HourVideo (Chandrasegaran et al., 2024).

| Group Size | Peak Allocated (MB) | 95th Percentile Allocated (MB) |
|---|---|---|
| Base Model | 15816.09 | — |
| 16 | 16718.12 | 16670.19 |
| 32 | 17349.22 | 17240.74 |
| 64 | 18606.21 | 18494.28 |

Table 4: Memory allocation statistics by group size.

95-percentile and peak of 1557.86 MB), QuickCodec workers have a peak RSS of 89.27 MB and 95th percentile of 88.68 MB per worker. The shared memory buffer is allocated with a constant size (Algorithm 2).

Table 4 shows GPU memory profile for a processing 30 minute video with different group sizes. We report both the peak allocated memory and 95th percentile allocated memory. The majority of memory is allocated for the base model, processing groups of size 16 and 32 increases peak memory utilization by 854.1 MB and 1533.13 MB, respectively.

## H    Implementation Details

QUICKCODEC, TorchCodec and Decord use FFmpeg to implement the decoder $\mathcal{D}$. QUICKCODEC uses system shared memory directly, instead of a parallelization library such as python's *multiprocessing*, as we find that serializing video tensors for inter-process communication introduces significant latency. Shared memory is designed for fast inter-process communication and is not secure. QUICKCODEC *should not be used on hosts running untrusted processes.*

## I    Limitations

As it is slow and resource intensive, most VideoLLMs are not trained to use their 1-2 FPS short video sampling rates when using processing long video (Bai et al., 2025; Chen et al., 2025a; Zhu et al., 2025a). Instead, they use very low sampling rates over large time-spans, as we discussed in Section 1. Therefore, VideoLLMs do not (yet) gain a large performance advantage by processing a large number of frames. However, it is clear that a model that has seconds-long gaps between frames can never capture fine-grained temporal and spatial details. Our hope is that making long video understanding (with realistic sampling rates) practical from a systems and algorithm perspective, we will empower the development of such models. Another limitation is that our QUICKCODEC timings only use H.264 coded video for timings. Although H.264 is the dominant standard, it is not universal.

