# OpenReview forum: "QuickVideo: Real-Time Long Video Understanding with System Algorithm Co-Design"
_TMLR — Accepted by TMLR_

### Review · Reviewer_bSCb · 2025-09-14

**Summary Of Contributions:**

This paper investigates a timely and important problem: enabling efficient long-video understanding with VLMs, where both video decoding and prefilling constitute severe bottlenecks. The authors propose QuickVideo, a system–algorithm co-design that integrates: (1) QuickCodec, a parallelized CPU-based decoder aligned with keyframes; (2) QuickPrefill, a grouped prefill and KV-cache pruning strategy; and (3) an overlapping execution scheme that overlaps CPU decoding with GPU inference. The method is validated on four video understanding benchmarks (VideoMME, LongVideoBench, LVBench, and MLVU) using Qwen2.5-VL-7B, achieving up to 3× end-to-end speedup with only minor accuracy degradation, thereby making long-video understanding more practical for real-world deployment.

Strengths:

1. The paper addresses a meaningful problem, optimizing video understanding, which is both necessary and has clear real-world applications.
2. The motivation is sound, as both prefilling and decoding present challenges; in particular, optimizing the codec process can significantly reduce the cost of transmitting large long-video files.

Weaknesses:

1. The paper lacks comparisons and discussion with related work on sparse attention for video LLMs[1,2].
2. Can this method be applied to accelerate video-LLM API serving? Do you have any test results in such scenarios?

[1] XAttention: Block Sparse Attention with Antidiagonal Scoring, ICML'25.

[2] MMInference: Accelerating Pre-filling for Long-Context Visual Language Models via Modality-Aware Permutation Sparse Attention, ICML'25.

**Audience:**

Yes

**Audience Explanation:**

The paper addresses a meaningful problem, optimizing video understanding, which is both necessary and has clear real-world applications.

**Claims And Evidence:**

Yes

**Claims Explanation:**

The motivation of this work is well-founded, and it is validated on four video understanding benchmarks (VideoMME, LongVideoBench, LVBench, and MLVU) using Qwen2.5-VL-7B.

**Requested Changes:**

1. Could you add comparisons and a discussion of related work on sparse attention for video LLMs [1,2]?
2. Can this method be applied to accelerate video-LLM API serving? Do you have any test results in such scenarios?

[1] XAttention: Block Sparse Attention with Antidiagonal Scoring, ICML'25.

[2] MMInference: Accelerating Pre-filling for Long-Context Visual Language Models via Modality-Aware Permutation Sparse Attention, ICML'25.

---

> ### Author Response · Authors · 2025-10-30
> **Rebuttal by Authors**
>
> **Q1**
> > Could you add comparisons and a discussion of related work on sparse attention for video LLMs [1,2]?
>
> Our QuickVideo framework complements and extends these efforts in several key ways. First, QuickVideo provides a training-free, plug-and-play solution that can be readily applied to existing VideoLLMs. Second, our QuickCodec component addresses the often-overlooked video decoding bottleneck, which becomes increasingly critical as video lengths extend to hours. Third, our overlapping execution scheme uniquely couples CPU-based video decoding with GPU-based sparse attention, achieving near-complete hardware utilization that is essential for real-time applications.
> Our method is orthogonal to previous prefilling methods for the CPU-GPU interleaving part.
>
>
> ---
>
> **Q2**
> > Can this method be applied to accelerate video-LLM API serving? Do you have any test results in such scenarios?
>
> No, for API serving, they usually receive the whole video at one time and we cannot manipulate how they process the video pre-filling externally. However, if we are able to modify how the system behind the API process the video, then we can apply our method there and thus to do the KV-caching pruning stuff.

---

### Review · Reviewer_iAq4 · 2025-09-15

**Summary Of Contributions:**

The paper targets long-video VideoLLM latency by designing QuickCodec, keyframe-aligned interval parallel decoding on CPU; QuickPrefill, grouped prefill with KV-cache pruning on GPU. It is a CPU/GPU overlap scheme. QuickVideo reports 2–3× faster loading at 16–32 cores and an end-to-end reduction from 69.7s to 20.0s on a 30-min, 1 FPS input (overlapped pipeline). It shows small average accuracy drops at moderate pruning ratios across VideoMME, LongVideoBench, LVBench, and MLVU.

Strengths:

1. The proposed method makes the pruning happening during the Prefill stage itself, while traditional methods mostly apply pruning during the decoding stage, since for long videos, prefill itself is the bottleneck.

2. The number of seek operations is reduced from being on the same order as the number of frames to just once per core, which is a targeted optimization for long video data-loading scenarios.

Weakness:

1. This paper of system-algorithm co-design is more like a known idea on multi-core decoding from an engineering perspective, but customized it as a decoder specialized for Video-LLMs. It would be helpful to have brief intuitions about model's choice, to make it more transparent to understand why the model is working, like the intuition why “key-norm (small)” works the best.

2. A concise clarification of Eq. (3) stating whether it is a bound/approximation and under what producer–consumer assumptions would make the analysis more robust.

**Audience:**

Yes

**Audience Explanation:**

1. CPU decoding tailored to low-FPS VideoLLM sampling + grouped prefill + overlap is well-motivated and practically useful.

2. The latency breakdown is informative.

3. QuickVideo does group-based prefill + KV pruning right away, discarding less important KV entries before decoding even begins. It makes long-video prefill feasible on commodity GPUs, and reduces activation and KV memory simultaneously, while keeping >95% accuracy.

**Broader Impact Concerns:**

None.

**Claims And Evidence:**

Yes

**Claims Explanation:**

1. QuickCodec scales to 64 cores and is consistently faster than common libraries at 16–32 cores; speed gains grow with video length. The number of seek operations is reduced from being on the same order as the number of frames to just once per core, eliminating much of the fixed overhead from seeking and resetting the decoder state. Within each segment decoding is still sequential, but across segments it can be parallelized over 16–64 cores. In practice, on common 16–32 core setups, this achieves a 2–3× speedup for a one-hour video, saving more than 20 seconds of loading time; the longer the video, the greater the advantage.

2. Long videos explode in prefill, while decoding is short. So QuickVideo shifts KV pruning into the prefill stage, which makes sense. Using a single heuristic (key-norm) plus grouping retains ~95–99% accuracy at ρ=0.5 on multiple long-video benchmarks, with an ablation over group size/ρ.

3. The method works better for longer videos. It requires a sufficient number of keyframes and longer videos to “sustain” parallelism. For very short videos or extremely sparse random access, the traditional “frame-by-frame seek” approach may actually be more suitable (the paper’s appendix discusses such cases).

**Requested Changes:**

1. QuickPrefill adopts the “key-norm (small)” rule as the default importance function. While this choice performs well empirically, the underlying motivation is only briefly discussed. Because retaining low-norm keys may be non-obvious at first glance (one might expect low-norm keys to correlate with lower attention), it is exciting to see more justification and can make the choice more convincing.

2. The paper provides a clear theoretical activation analysis and latency breakdown, but does not include synchronized CPU/GPU memory–time profiles. Since interval-parallel decoding may increase CPU RSS and pinned buffers as thread/interval counts grow (due to decoder state and temporarily decoded frames), it would be valuable to confirm whether CPU memory remains manageable in longer videos at higher parallelism. Consider including synchronized memory-time profiles (CPU RSS, pinned host memory, GPU allocated/KV/activations) for representative settings (e.g., 30–60 min videos at 1–2 FPS; $c\in\{16,32,64\}$; $\rho\in\{1.0,0.5,0.2\}$; a couple of group sizes). Peak and 95-percentile summaries would suffice and would complement the latency breakdown.

3.  Eq. (3) models total latency as $\max(t_{\text{dec}}+t^g_{\text{prefill}},\,t_{\text{prefill}}+t^g_{\text{dec}}+\Delta)$, which seems to mainly reflect boundary groups. Since the time for prefilling and decoding in each group varies, the steady-state duration may be better approximated by $\sum_{g=1}^G \max(t^g_{\text{dec}},t^g_{\text{prefill}})$ plus startup/cleanup. A short clarification on whether Eq. (3) is intended as a bound, an approximation, or an empirical fit would strengthen the presentation.

---

> ### Author Response · Authors · 2025-10-30
> **Rebuttal by Authors**
>
> **Q1**
>
> The key-norm (small) is not a simple heuristic, but sourced from previous findings, including paper [1][2], where we all find that the key-norm has a strong correlation with the computed attention scores. Later on, the Q-Filter paper [3] further explains the underlying reason of this phenomenon, give its mathematical explanation. We accidentally find this phenomenon before we realize that it has already been discovered by these papers, so we simply cite these papers as an explanation of why “key-norm (small)” works.
>
> ---
>
> **Q2**
>
> CPU memory profiles corresponding to figure 5: QuickVideo’s CPU memory is made up of 3 components: the torch main process, QuickCodec workers, and the shared memory block into which video frames are loaded (line 2 in Algorithm 2). The torch main process has consistent memory utilization (RSS 95-percentile and peak of 1557.86 MB), QuickCodec workers have a peak RSS of 89.27 MB and 95th percentile of 88.68 MB per worker. The size of the shared memory buffer is always $|\mathcal{I}| \times 3 \times h \times w$ bytes, where $|\mathcal{I}|$ is the number of frames loaded. GPU memory profiles: please refer to the table below for GPU memory profile (corresponding to the latency experiment in figure 5) for a processing 30 minute video with different group sizes.
>
>
> | Group Size | Peak Allocated (MB) | 95th Percentile Allocated (MB) |
> |-------------|---------------------|--------------------------------|
> | Base Model  | 15816.09            | —                              |
> | 16          | 16718.12            | 16670.19                       |
> | 32          | 17349.22            | 17240.74                       |
> | 64          | 18606.21            | 18494.28                       |
>
> The majority of memory is allocated for the base model, processing groups of size 16 and 32 increases peak memory utilization by 854.1 MB and 1533.13 MB, respectively. We will update the manuscript to include memory profile results corresponding to latency experiments.
>
> ---
>
> **Q3**
>
> We agree that the sum presentation is more clear than the boundary formula. Eq. (3) is designed to be an empirical model for latency, which we measure in 4.3. We will update the manuscript with the clearer formulation for Eq. (3) and clarify its motivation.
>
> #### References
>
> [1] Devoto, Alessio, Yu Zhao, Simone Scardapane and Pasquale Minervini. “A Simple and Effective L\_2 Norm-Based Strategy for KV Cache Compression.” ArXiv abs/2406.11430 (2024): n. Pag.
>
> [2] Wen, Zichen, Yifeng Gao, Shaobo Wang, Junyuan Zhang, Qintong Zhang, Weijia Li, Conghui He and Linfeng Zhang. “Stop Looking for Important Tokens in Multimodal Language Models: Duplication Matters More.” ArXiv abs/2502.11494 (2025): n. Pag.
>
> [3] Godey, Nathan, Alessio Devoto, Yu Zhao, Simone Scardapane, Pasquale Minervini, Eric Villemonte de la Clergerie and Benoît Sagot. “Q-Filters: Leveraging QK Geometry for Efficient KV Cache Compression.” (2025).

---

### Review · Reviewer_Wskh · 2025-10-16

**Summary Of Contributions:**

## **Summary**

This paper presents **QuickVideo**, a system–algorithm co-design framework for real-time long video understanding in **Video Large Language Models (Video LLMs)**. The work tackles two major bottlenecks in current long-video inference stack (1) **sequential video decoding** and (2) **memory-intensive prefilling**, and proposes three solutions:

- **QuickCodec**, a parallelized, keyframe-aligned video decoder achieving **2–3× speedup for hour-long videos** over existing tools such as Decord and TorchCodec.
- **QuickPrefill**, a group-based prefilling approach with **KV cache pruning** that significantly reduces GPU memory usage with **less than 3% accuracy degradation**.
- An **overlapped CPU–GPU execution scheme** that overlaps decoding and inference, **reducing total inference time for a 30-minute video from 69.7 s to 20.0 s** (more than **3× speedup**).

---

## **Strengths**

- Clear system–algorithm co-design that bridges efficiency and practicality for long video inference.
- Strong empirical validation with multiple long-video benchmarks.
- Effective demonstration of real-time capabilities with measurable latency reductions and memory savings.

---

## **Weaknesses**

- **HourVideo benchmark results are missing.** Since HourVideo was designed specifically for hour-long video-language understanding, including it would help contextualize performance even if the numbers are less favorable.
- **Comparison against token pruning methods** for KV cache compression is missing. Including this would help illustrate the Pareto frontier of speed versus accuracy and benefit the community.  You may consider a recent work like this: https://research.nvidia.com/labs/lpr/storm/
- In **Sec. 4.2 (Table 3)**, it is unclear whether the **attention matrix** is materialized when computing attention-score-based KV cache pruning. If so, this could be prohibitively expensive for GPU memory; clarification is needed.
- The evaluation uses only a **single model**. Adding one more model (e.g., from a different family such as **LLaVA-Video**) on 1–2 benchmarks would help confirm that performance trends in Table 1 generalize across models.
- **TorchCodec evaluation details.** TorchCodec supports both CPU and CUDA backends. Please clarify whether the reported comparisons use the CPU or GPU version. If only CPU results are reported, consider including **TorchCodec[CUDA]** benchmarks for completeness since QuickVideo also has meaningful implications for improving video LLM training pipelines.
- **Conduct benchmarks on newer GPU hardware** such as **H100 or B200** to contextualize real-time performance claims on modern GPUs.

**Overall:** This is a well-executed paper that advances the field of efficient long video understanding through thoughtful system–algorithm design.

**Audience:**

Yes

**Audience Explanation:**

Yes. The work will interest researchers in multimodal learning, efficient inference, and video understanding. It provides practical system-level improvements for VideoLLMs, a topic gaining traction in both academic and applied machine learning communities.

**Broader Impact Concerns:**

The paper already includes a **Broader Impact Statement** that appropriately discusses both accessibility benefits and potential privacy risks.

**Claims And Evidence:**

Yes

**Claims Explanation:**

The summary already covers the main evidence. The claims are generally well supported by experiments and benchmarks, but addressing the identified weaknesses such as including HourVideo results, comparisons to token pruning, and clarification on TorchCodec CPU vs. GPU results would make the evidence more convincing.

**Requested Changes:**

See weaknesses above. Addressing those points, particularly adding HourVideo results, token pruning comparisons, clarification on attention matrix computation, one additional model evaluation, clarification on TorchCodec CPU vs. GPU results (and including TorchCodec[CUDA] benchmarks if possible), and conducting benchmarks on newer GPU hardware such as H100 or B200 to contextualize real-time performance claims, would strengthen the submission.

---

> ### Author Response · Authors · 2025-10-30
> **Rebuttal by Authors (Part 1)**
>
> **Q1**
> > HourVideo benchmark results are missing
>
> HourVideo is indeed one of the representative benchmarks that focus on hour-long video understanding. We also tried to evaluate on HourVideo, which is built on videos from Ego4D. Currently we are still waiting for the approval licence for downloading Ego4D video datasets and thus this has prevented us from conducting evaluation for now. We would definitely like to evaluate our method on more hour-level video understanding benchmarks if everything goes well.
>
> However, we would like to state that our current included benchmarks also contains a number of hour-level videos with associated questions (See table below). For example, VideoMME-Long subset has an average video length of 2466 seconds, and MVLU and LVBench both have videos even longer near 2 hours. Therefore, we do believe that existing benchmarks are enough for us to clarify the effectiveness of our methods.
>
> | Benchmark Name | Ave Duration (s) | Max Duration (s) |
> |----------------|------------------|------------------|
> | VideoMME       |             1017 |             3600 |
> | LongVideoBench |              473 |             3600 |
> | MLVU           |              930 |             7200 |
> | LVBench        |             4101 | -                |
>
> ---
>
> **Q2**
> > Comparison against token pruning methods
>
> Thanks for the suggestion for adding Storm as a baseline. We read the paper and found that Strom is actually a training-needed method and it’s also associated with some custom mamba architecture to do the temporal pooling operation. While we appreciate Strom’s impressive performance on long video understanding and inference efficiency, it’s not quite possible to directly compare our pruning method, which is training-free, to Strom.
>
> However, the 3 operation mentioned in the paper: spatial pooling, temporal pooling and temporal sampling is indeed a universal operation that we may be able to implement in our pipeline, as long as we replace method used in temporal pooling from mamba to a simple averaging operation. We implemented the temporal pooling operation on the visual embeddings before feeding them into the language model part. And the results are as follows:
> | Num Frames | average pooling factor | VideoMME |
> |------------|------------------------|----------|
> |       1024 |                      1 |       62 |
> |       1024 |                      4 |    16.74 |
> |         32 |                      1 |    62.41 |
> |         32 |                      2 |    57.81 |
> |         32 |                      4 |    52.74 |
>
> Clearly, we can see there is a clear drop by applying the average pooling without any fine-tuning for both 1024 and 32 frames setting, and does not work well compared to the key-norm (small) method.
>
> ---
>
> **Q3**
> > it is unclear whether the attention matrix is materialized
>
> The attention score method is just a baseline, we indeed materialized that attention matrix when computing. Since Flash-Attention cannot return the full attention score matrix, we have to rollback to the pytorch naive attention computation to get the attention scores, which can be both more computation and memory intensive than flash-attention.
>
> However, the key-norm based KV-cache selection method can coordinate well with Flash-Attention without the need of the full attention score matrix, therefore it can benefit from both the efficiency of flash-attention and higher KV cache pruning performance. We use the key-norm based method as the default method of QuickVideo.
>
> We will add corresponding clarification for it in the final version of the paper.
>
> ---
>
> **Q4**
> > The evaluation uses only a single model.
>
> Adding more models is definitely a todo for the future work. We will put results of these models in the paper once the codes for new custom model is ready.
>
> ---
>
> **Q5**
> > TorchCodec evaluation details.
>
> Our benchmarks use the cpu backend for TorchCodec. We have not been able to use/compare against the CUDA backend due an ongoing issue in torchcodec CUDA builds (see [1](https://github.com/meta-pytorch/torchcodec/issues/912), [2](https://github.com/meta-pytorch/torchcodec/issues/998)). We are working to support NVDEC acceleration for QuickVideo and we will hopefully be able to compare it to TorchCodec[cuda] in the future. We will clarify the manuscript to specify that we are using TorchCodec[cpu].

---

> > ### Author Response · Authors · 2025-10-30
> > **Rebuttal by Authors (Part 2)**
> >
> > **Q6**
> > > Conduct benchmarks on newer GPU hardware such as H100 or B200
> >
> > We have recreated the timings in figure 5 using an H100 SXM GPU and AMC Ryzen 9 7950x CPU.
> > | Stage (seconds)                    | Baseline | QuickVideo | QuickVideo (Interleaved) |
> > |----------------------------|-----------|-------------|---------------------------|
> > | Video Loading               | 24.9937   | 15.3555     | 5.4760                    |
> > | Prefill                | 7.3934    | 7.3827      | 10.3104                   |
> > | Total | 32.3871   | 22.7383     | 15.7864                   |
> >
> > Interestingly, we observe that on the much faster H100 GPU, the prefill stage becomes bottlenecked by video loading. Not all video loading latency can be “hidden” behind GPU prefill, resulting in the interleaved GPU prefill stage being marginally slower. Results are averaged over 5 runs each.

---

### Decision · Action_Editor_sKZG · 2025-12-22

**Recommendation:** Accept with minor revision

**Additional Comments:**

Both Reviewer iAq4 and Wskh requested the results of HourVideo, as the benchmark was specifically designed for evaluating long-video visual-language understanding. In a journal paper, it's important to provide a clear context for where the method stands. Therefore, it's important to evaluate the HourVideo dataset. As the reviewer mentioned, "even if the numbers are less favorable". The reviewer iAq4 also mentioned that they would reconsider their negative rating if the authors could incorporate all the changes.

In the rebuttal period, the authors stated that they could not evaluate the HourVideo because they are awaiting access to the Ego4D video. As we now have more time until the camera-ready deadline, the authors should have time to obtain data access and provide the results.

Considering both the strength/weakness outlined by reviewers, the AE recommends "Accept with minor revision".

In the revised paper, please include the HourVideo results for reference and integrate the reviewers' requested changes/experiments into the main paper.

**Audience:**

Yes

**Audience Explanation:**

Yes, TMLR's audience interested in video understanding, multimodal learning, and system-algorithm co-design would be interested in this paper's findings.

**Claims And Evidence:**

Yes

**Claims Explanation:**

Yes, the paper's key claims lie in accelerating long-video understanding with VideoVLM. Reviewers found the experiments solid, with strong results across multiple benchmarks (VideoMME, LongVideoBench, LVBench, and MLVU) using Qwen2.5-VL-7B, and demonstrated only small performance degradation.